# Research on Consumer Trust Mechanism in China's B2C E-Commerce Platform for Second-Hand Cars

**Xueqian Li \*, Jiaqi Ma, Xinyu Zhou and Ruixia Yuan**

Business School, University of Shanghai for Science and Technology, Shanghai 200093, China
\* Correspondence: xqli81@usst.edu.cn

**Abstract:** The rapid development of China's e-commerce industry has led to the rise of China's second-hand car e-commerce. With the increasingly rich trust theory and the rapid development of e-commerce platforms, the issue of online consumer trust has attracted more and more scholars' attention. This paper takes China's B2C second-hand car e-commerce platforms as an example, combines the second-hand car research conclusions and consumer trust theory, and conducts a systematic study on the formation of consumer trust in second-hand car e-commerce platforms. Based on the trust of individual consumers, system environment, website/APP and platform companies, this paper explores the influencing factors of consumer trust and constructs the influencing factors model of trust formation. The empirical study was conducted by using the structural equation model and multiple regression to verify the degree of fitting of the theoretical hypothesis and the model. The research results have a certain reference value for the development of second-hand car e-commerce platforms.

**Keywords:** second-hand car; e-commerce platforms; consumer trust; structural equations; multiple regression





## 1. Introduction

Second-hand car e-commerce is a new business model of China's second-hand car industry. In recent years, the development of second-hand car e-commerce in China has been particularly rapid. From the perspective of car ownership, Chinese automobiles are changing from incremental to stock, and the upgrade and replacement rate will increase year by year.

The trade volume has increased due to the trend of replacing second-hand cars. In 2021, China's second-hand car transaction volume exceeded CNY 1.1 trillion, with a year-on-year growth of 27.3%, becoming a veritable trillion-yuan market. Around 2013, China's second-hand car trading market began to see an investment boom, and investment institutions believed that e-commerce would become the main channel of second-hand car trading. Therefore, there was an emergence of a number of second-hand car e-commerce companies. Under the influence of the booming second-hand car market and the popularity of e-commerce, the number of second-hand cars traded on e-commerce platforms showed a rapid growth trend. In 2021, the transaction scale of second-hand e-commerce in China reached CNY 400.17 billion, with a year-on-year growth of 29.27%. However, compared with most other e-commerce platforms (comprehensive and vertical), the development speed of second-hand e-commerce is relatively stable and slow. Over the past three years, market penetration growth has fallen far short of expectations, as shown in Figure 1. What factors restrict its development in the end? Through the survey, the researchers found that the key obstacle to the development of China's second-hand car e-commerce platforms is still the low level of consumer trust. Compared with general commodities, second-hand cars have a high unit price, complex quality information, life safety and low consumption frequency, so there may be some unique mechanism in the process of consumer trust

formation. When the traditional trading mode of the market has not yet cultivated enough consumer trust, will people have high trust in the natural place? This is doubtful. Therefore, this is a research work with both theoretical and practical value to deeply understand the formation mechanism of consumer trust in the second-hand car trading scenario and propose measures to enhance consumer trust based on this.

**Figure 1.** China's second-hand car e-commerce permeability from 2019–2022.

The business model of a second-hand car e-commerce platform is divided into four types. The C2B mode starts from the C-end seller users, who sell the car to the B-end car dealers; such platforms include Che Zhibao, Da Souche, etc. The B2C mode leads the source of B-end vehicles to the demand of the C-end. Such platforms include Youxin Second-hand Car, 99 Good car, etc. B2B and C2C businesses directly face two car dealers or two major trading user groups. Platforms focusing on a B2B model include Youxin Pai, Cheyipai, etc., while those focusing on C2C model include person-to-person car, Guazi second-hand car, etc. In general, after years of exploration and operation, brand value, operation experience and business channels are the core barriers of competition for each e-commerce platform.

In these four types of businesses, the highest degree of marketization is in the B2C and C2C modes, that is, the end-user-oriented businesses. We want to explore the transaction process facing end users and the trust formation mechanism of end consumers in the transaction process. Since the B2C model is a transaction between enterprises and individuals, while the C2C business model is essentially a transaction between individuals, their transaction processes and the formation mechanism of consumer trust may be very different. Therefore, it is necessary for this paper to choose one of the modes to study. Considering that the B2C model has developed well in China's second-hand car e-commerce market and is more likely to become the mainstream model, we decide to take the B2C model as the research scope of this paper.

The research on the second-hand car market originated from Akerlof's "The Market for Lemons: Quality Uncertainties and the Market Mechanism" published in 1970 [1]. He discussed and analyzed three industries with asymmetric information, including the analysis of lemon market problems caused by the second-hand car market, and pointed out that the reason why there are problems in the second-hand car market is that the seller has more information than the buyer. Information asymmetry is an important aspect that affects consumer trust. As a critical factor for the success of online enterprises, e-services,

trust has been proven to have a significant impact on customer behavioral intentions (Lee and Lee, 2005; Tajvidi et al., 2017), even in the context of second-hand cars [2,3].

Trust has been widely studied and considered as the driving factor of e-commerce (Alalwan et al., 2017) [4], but there is still relatively little attention to revealing how trust develops and its role in the second-hand car business. The unpredictable environment due to the lack of face-to-face communication is the main reason why previous e-commerce/s-commerce studies have explored the mechanism of trust formation (Gefen and Straub, 2003; Kim and Park, 2013) [5,6]. Factors such as social support, information quality and social presence (Chen and Shen, 2015; El Amri and Akrout, 2020) have been proved to be able to help sellers gain customers' trust [7,8]. However, there are huge differences between second-hand cars and general goods. Whether the mechanism of trust has changed and how it affects follow-up behavior is still unknown in the e-commerce of second-hand cars. We believe that business characteristics, product attributes and other factors can affect consumer trust, and website authentication can also promote consumer trust (Hillman, 2017) [9]. When consumer perception is consistent with product quality, consumer satisfaction will increase, and thus trust in the seller will also increase (Fang et al., 2014) [10]. Consumers have higher confidence in products with a quality certification or quality endorsement. When the transaction is at risk, consumers are willing to believe in policies and laws, and this effective governance can significantly improve consumer trust (Hartl et al., 2016) [11].

The existing research tends to regard trust as an aggregation structure (Wan et al., 2016; Jiang et al., 2019) [12,13]. The whole concept is problematic because the customer's behavior intention depends not only on the expectation of the product but also on their attitude towards the person who provides them with services (Kim and Park, 2013) and the service level of the service provider [14]. Jone (2014) takes perceived website quality, third-party authentication and network security performance as factors affecting e-commerce consumer trust [15]. In addition, personal experience factors cannot be ignored. Sutanonpaiboon (2008) divided the trust dimensions that directly affect consumers' purchases into media trust and trust in individual characteristics, including individual ability, kindness and honesty [16]. However, there is little research on how to form trust in the Internet market of second-hand cars. The relevant literature still lacks a micro-analysis of trust mechanisms and a deeper understanding of how they affect customer behavior. Therefore, in the context of second-hand car Internet platforms, we distinguish trust at the individual level, platform level and system level. These sources of trust are both different and interrelated. What is the internal relationship between them and the formation of consumer trust? This is the problem that we need to solve in our work. This paper has made some contributions in theory, mainly as follows: (1) Based on the consumption decision-making process, the consumer trust mechanism model has been constructed from the three levels of system, enterprise and individual, which is a beneficial extension of consumer trust theory; (2) Second-hand car e-commerce is a special type of e-commerce, which has developed rapidly in China. However, there has been no research on this issue in the past, so this paper has enriched the theoretical research of e-commerce.

This paper is organized as follows: Firstly, it analyzes the market background and research basis of the research. Section 2 establishes a theoretical model and puts forward assumptions. Section 3 conducts empirical analysis. Finally, it summarizes the research conclusions and emphasizes the management implications for enterprises.

## 2. Materials and Methods

### 2.1. Research Model

Moorman (1993) first proposed a three-stage model of network trust: trust influencing factors, trust generating process and trust outcomes [17]. Mayer's (1995) outcome model assumes that the characteristics of both the trusted party (competence, goodwill and honesty) and the trusting party (propensity to trust) have an impact on overall trust [18].

Later, Gefen and Straub (2004) extended the trust factor model [19]. The main research hypothesis is proposed based on the reading study of relevant literature.

### 2.1.1. Consumer Personal Factors

Personal factors are important aspects affecting trust formation, and personal factors include three dimensions: personal inherent trust tendency, personal vehicle knowledge level and personal online shopping experience.

1. Personal inherent trust tendency

Personal inherent trust tendency arises from the individual's past life experiences.

Lee (2001) found that trust tendency directly affects personal online trust [20]. Heyns and Rothmann (2015) concluded that a high level of personal inherent trust tendency would promote perceived trustworthiness [21]. The following hypothesis is proposed:

**H1.** *Consumers' personal inherent trust tendency fosters the consumer trust in B2C second-hand car e-commerce platforms.*

2. Personal vehicle knowledge level

In the transaction situation, the degree of consumer knowledge of a good can directly affect trust formation.

Luhmann (1979) believed that the more familiar consumers are, the easier it is to alleviate internal doubts and confusion, and thus the more likely they are to develop trust [22]. Wu Shaowei (2006) argued that the size of a consumer's knowledge base affects his ability to receive, process and perceive information [23]. All these indicate the important role of knowledge level for the formation of consumer trust.

For a commodity with many technical parameters such as automobiles, this paper argues that consumers who have more knowledge about vehicles are less likely to reject the second-hand car e-commerce transaction method. The following hypothesis is proposed:

**H2.** *Consumers' personal vehicle knowledge level increases the consumer trust in B2C second-hand car e-commerce platforms.*

3. Personal online shopping experience

Koufaris (2002) believed that consumers with more experience can better use online shopping as a consumption channel [24]. Chen Chen (2015) stated that those consumers who have experience in online shopping have a deeper understanding of e-commerce and are more receptive to new models [25]. This paper argues that consumers' previous online shopping experience can contribute to whether they choose to buy second-hand cars online. The following hypothesis is proposed:

**H3.** *Consumers' personal online shopping experience raises the consumer trust in B2C second-hand car e-commerce platforms.*

### 2.1.2. System Environmental Factors

In addition to individual subjective factors, objective environmental factors are also the main aspects affecting trust formation. The objective environment of second-hand car e-commerce transactions includes laws and regulations; third-party certification; and industry technology.

1. Laws and regulations

McKnight et al. (2000) believe that sound laws and regulations will affect consumers' trust level [26]. Chen Yini (2010) argued that a secure environment brings behavioral dependence to consumers [27]. There are many legal risks in second-hand car transactions, such as stolen cars, illegal unprocessed cars and concealed vehicle failure history. Therefore, the laws and regulations related to second-hand car network transactions are particularly important. It directly determines the consumer's sense of security, which in turn affects consumer trust. The following hypothesis is proposed:

**H4.** *Laws and regulations influence the consumer trust in B2C second-hand car e-commerce platforms.*

2. Third-party certification

The third-party certification mechanism is the basic condition for the explosive growth of e-commerce in China. For example, Taobao sets up crown certification for sellers and the 268V vehicle inspection service is provided by the CarEasy platform. Chen Xianyou (2013) believed that third-party certification would improve consumers' trust [28]. Yoon and Occeña (2015) pointed out that third-party certification significantly affects consumers' trust in e-commerce platforms [29]. Vehicle transactions involve a large number of technical parameters, which are a key basis for second-hand car valuation and consumer purchase decisions. Authoritative certifications make consumers more likely to trust e-commerce platforms. The following hypothesis is proposed:

**H5.** *Third-party certification of platforms promotes the consumer trust in B2C second-hand car e-commerce platforms.*

3. Industry technology level

Yan Zhonghua (2004) argued that technological trust is a subjective concept that facilitates transactions [30]. Wang Shouzhong (2007) pointed out that the Internet has been able to develop in China, which is inseparable from the developed financial system and logistics and distribution system, which are closely related to transactions [31]. Tian Zhaohui (2020) argued that second-hand car e-commerce platforms should facilitate transactions with the help of innovative technologies such as artificial intelligence to accomplish services such as product pricing, intelligent inspection and integration of logistics information [32]. The level of industrial technology referred to in this study is mainly that of the two industries, automotive and Internet. The following hypothesis is proposed:

**H6.** *Industry technology level may increase the consumer trust in B2C second-hand car e-commerce platforms.*

### 2.1.3. Website/APP factors

At the first step of forming online trust between the two sides of the transaction, those at both sides of the transaction have never met each other and are strangers to each other; the first impression or some details will affect the formation of trust.

1. Ease of use of the website/APP

Ease of use is essentially the cost of use for consumers. Stanford (2002) argued that a website that is designed to have consumers up and running quickly gives a good start to building consumer trust [33]. Zeng Guichuan (2015) showed that the ease of use of a mobile app significantly affects the user's usage behavior [34]. In this paper, we argue that the ease of use of the website/APP of second-hand car e-commerce platforms in terms of appearance design, navigation functions and related links affects consumers' perceptions and experiences. The following hypothesis is proposed:

**H7.** *The ease of use of the website/APP increases the consumer trust in the B2C second-hand car e-commerce platform.*

2. Security of website/APP

The security of a website/APP includes two aspects: transaction security and personal information security. Zhang Gaoliang (2014) showed that the level of security of online transactions has a significant positive impact on users' trust and satisfaction [35]. The importance of website/app security is accentuated by the high customer unit price of second-hand car transactions. The following hypothesis is proposed:

**H8.** *The security of a website/APP fosters the consumer trust in a B2C second-hand car e-commerce platform.*

#### 2.1.4. Platform Company Factors

The background, reputation and level of operation of the e-commerce platform company in the B2C model are also factors that consumers consider.

1. Company size

The company size means the depth of the company's product offerings and the breadth of its services. Liu Junqing (2015) showed that the larger the e-commerce platform, the higher the level of trust generated [36]. Lv Xiaojing (2019) believes that the larger scale of e-commerce platforms means that they can provide better services, and they can better meet the needs of different consumers [37].

Larger second-hand car e-commerce platform companies generate stronger network externalities and attract more consumers' attention. Additionally, they have better second-hand car sources as well as supporting service providers, which further facilitate consumer experience and help generate trust. Therefore, the following hypothesis is proposed:

**H9.** *The strength of the platform company's size increases the consumer trust in a B2C second-hand car e-commerce platform.*

2. Company brand reputation

McKnight et al. (1998) argued that corporate reputation is a key factor for e-commerce merchants to build trust [38]. Xiang Luquan (2020) pointed out that the better a company's brand reputation, the more it can improve consumers' trust in the merchant [39]. The following hypothesis is proposed:

**H10.** *The brand reputation of the platform company increases the consumer trust in a B2C second-hand car e-commerce platform.*

Based on the research hypotheses established above, the research model shown in Figure 2 was constructed.

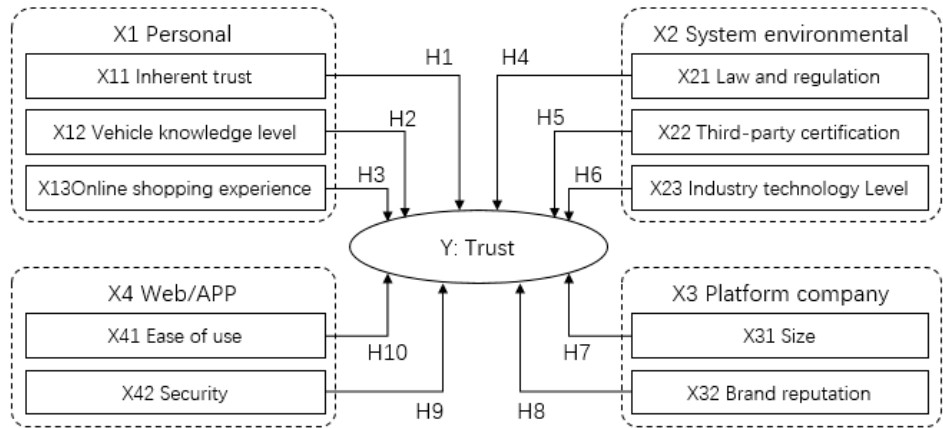

**Figure 2.** Consumer trust mechanism model of second-hand car e-commerce platforms.

#### 2.2. Research Methodology

This study collected data by questionnaire. Based on the relevant literature and the consumer trust characteristics of a used-car e-commerce platform, the scale was developed and more than three questions were designed for each measurement variable to improve the validity of the questionnaire, as shown in Table 1. After the design and pre-test of the first draft of the questionnaire, the questionnaire that meets the needs of this study was finally formed.

**Table 1.** Variables and index.

| Variables | | Index |
|---|---|---|
| Personal trust tendency | PT1 | I believe that in most cases, human nature is kind. |
| | PT2 | Generally speaking, I usually trust a person or thing, even if I don't know much about it. |
| | PT3 | I will not take the initiative to suspect a person until there is evidence that this person is not trustworthy. |
| Personal vehicle knowledge | PK1 | I have the experience of buying vehicles and have a certain understanding of the nature of vehicles. |
| | PK2 | I understand the basic knowledge of vehicles, such as engine, gearbox, chassis, etc. |
| | PK3 | I understand the meaning of some vehicle performance parameters and can compare them, such as horsepower, torque, fuel consumption, etc. |
| Personal online shopping experience | PE1 | When I have shopping needs, I will choose to use e-commerce to find goods and have done online shopping. |
| | PE2 | I understand the process of online shopping, have dealt with customer service, and have tried after-sales service. |
| | PE3 | During online shopping, I can smoothly complete the steps of product retrieval, order confirmation, online payment, etc. |
| Laws and regulations | LR1 | I know more about the existing laws and regulations related to e-commerce industry and automobile industry in China |
| | LR2 | I think the relevant e-commerce laws can effectively ensure the safety of online shopping. |
| | LR3 | I think the relevant laws and regulations of the second-hand car industry can guarantee the rights and obligations of both parties. |
| Third-party certification | TR1 | I think the third-party certification is very important for the used car e-commerce platform. |
| | TR2 | I think the third-party authoritative certification provides guarantee for the capability and legal compliance of the second-hand car e-commerce platform. |
| | TR3 | I believe that a third party can protect my rights and interests when the other party does not act in good faith. |
| Industrial technology | IT1 | I usually pay attention to the development of some industrial technologies and future technologies. |
| | IT2 | I think the technology of automobile industry can guarantee the quality of automobiles and meet my needs. |
| | IT3 | I think the Internet night technology can enable me to use the Internet safely and efficiently. |
| Website/APP perceived ease of use | PU1 | The rationality of website interface design or APP interactive interface design helps me believe that the website can be used for the sake of users. |
| | PU2 | The timeliness of the response of the website or APP helps me believe in the authenticity of the platform's information. |
| | PU3 | When browsing the website or APP, when I encounter problems, if I can find official guidance on the website APP, I will feel very convenient. |
| Website/APP perceived security | PS1 | The authority and popularity of the certification party and partner of the website or APP will affect my judgment on the security of the website. |
| | PS2 | When browsing the website or APP, if the website or APP will inform the user of the personal information protection policy, then I think it is safer to change the website. |
| | PS3 | When confirming the order and paying, if it is a traditional payment method (WeChat, Alipay, UnionPay card, etc.), I will feel that the transaction payment is safer. |

**Table 1.** *Cont.*

| Variables | Index | |
|---|---|---|
| company size | CS1 | I think large companies can provide me with better products and services. |
| | CS2 | I think a strong company can better meet my needs. |
| | CS3 | If the brand awareness of the two companies is equal, I will choose the company with a larger market share. |
| Company brand reputation | CR1 | I think companies with good reputation can be more honest and trustworthy with consumers. |
| | CR2 | I think companies with high brand awareness pay more attention to consumer relationship maintenance. |
| | CR3 | If both companies can meet my needs, I will choose the company with less negative comments. |
| Consumer trust | CT | I believe that the company has the intention and ability to provide products and services that satisfy me. |

First, we interviewed second-hand car consumers and second-hand car experts, combined with existing theoretical scales, for the initial development of the questionnaire. It was distributed to neighboring friends as a trial fill and revised and finalized. At the same time, before we asked the respondents to fill in the questionnaire, we solicited the opinions of the respondents and explained the purpose of the questionnaire in detail. The questionnaire includes an introduction and body. In the introduction, we explained the purpose and significance of this study in detail, and emphasized that it is only for theoretical research, does not involve personal privacy and does not pose any risk to individuals and the public. The respondents could refuse to fill in the questionnaire. In the body, they did not need to fill in the identity information. Finally, we used the "Questionnaire Star" to make a questionnaire that is easy to fill in, and distributed it using online tools. Then, it was distributed to them using web tools, and, finally, 468 questionnaires were collected, of which 410 were valid.

After statistical analysis, it was found that 57.56% of the respondents were female and 42.44% were male, basically in line with the reality that women are the main group in e-commerce consumption. From the age distribution, it can be seen that 20–29 year-old plus 30–39 year-old people accounted for 69.27% of the respondents, and the younger group showed a stronger interest in new things. People with a bachelor's degree or above accounted for the majority, accounting for 67.07%.

The number of people who have had the experience of purchasing a car in the questionnaire survey is 237, accounting for 57.80%, more than those who have not had the experience of purchasing a car. The number of respondents who have a license is 68.78%, more than those who do not have a driver's license, basically in line with the proportion of people who have a driver's license in the total population. A total of 92.44% of the respondents maintain the habit of online shopping; only 7.56% said they do not shop online at all. People who have used a second-hand trading platform accounted for 39.27% of the interviewed population, of which only 7.56% have used a second-hand car e-commerce platform, indicating the low popularity of second-hand car e-commerce platforms. It can be found that the basic background of the interviewees is basically in line with the development pattern of Chinese society and the e-commerce industry, which lays the foundation for the subsequent analysis and research.

## 3. Empirical Analysis

### 3.1. Reliability and Validity Analysis

In the reliability analysis, the Cronbach's alpha coefficient of the measured questionnaire data is 0.919, which is greater than 0.8, indicating that the overall reliability of the questionnaire is very good. In the validity analysis, the value of sample KMO is 0.927,



which is higher than 0.7, indicating that there are more common factors among variables. The Bartlett's test of sphericity approximate chi-square value is 7755.876, the degree of freedom reaches 528 and the obtained result is significant, indicating that there are common factors in the correlation matrix of the group, based on which this questionnaire is considered suitable for factor analysis and the total amount reaches validity.

### *3.2. Model Test*

### 3.2.1. Validation of the Structural Equation Model

In this study, the model was tested for goodness of fit using AMOS, and the chi-square test index of the model goodness of fit reached $\chi^2/df$ 2.019, an RMSEA of 0.05, a TLI index of 0.928, a CFI index of 0.940, a PNFI index of 0.741 and a PGFI index of 0.783; all of these major indices reached a very high level. The GFI value is 0.887, the AGFI value is 0.856 and the NFI value is 0.889, and the regression levels of these three indices are within a reasonable range. See Table 2 for details. Based on the above indices, we can assume that the fit of the model can meet the needs of validation.

**Table 2.** Analysis of the results of the overall model fit test.

| Indicator | Evaluation Indicators | | Value | Judgment |
| | Able to Accept | Better | | |
|---|---|---|---|---|
| $\chi^2/df$ | 3~5 | 1~3 | 2.019 | Better |
| RMSEA | 0.08~0.10 | <0.08 | 0.050 | Better |
| GFI | 0.80~0.90 | >0.90 | 0.887 | Able to accept |
| AGFI | | | 0.856 | Able to accept |
| NFI | | | 0.889 | Able to accept |
| TLI | 0.80~0.90 | >0.90 | 0.928 | Better |
| CFI | | | 0.940 | Better |
| PNFI | >0.50 | | 0.741 | Better |
| PGFI | | | 0.783 | Better |

In this study, the structural equation was chosen to test the paths of the two groups of consumer personal factors and system environmental factors; SPSS was chosen to conduct regression analysis on the website/APP factors and the platform company factors. The reason for this is that both consumer personal factors and system environmental factors contain three independent variables, which can be correlated two by two in the structural equation; both website/APP factors and platform company factors contain two independent variables, and if the structural equation is chosen to describe their paths, it will affect the overall fit and the quality of the path test. Based on this, the variables are divided into two groups and, using two methods, are tested below.

Using AMOS software, the degree of fit of the consumer trust part of the model was calculated by choosing the maximum likelihood method and the oblique rotation method, and the results are shown in Table 3.

**Table 3.** Table of the results of the overall goodness-of-fit test of the study model.

| Indicators | $\chi^2/df$ | GFI | AGFI | CFI | TLI | RMSEA |
|---|---|---|---|---|---|---|
| Results | 3.168 | 0.893 | 0.860 | 0.895 | 0.875 | 0.073 |
| Judgment | True | True | True | True | True | True |

The results of the path test for a total of six variables in two groups of factors, consumer personal factors and system environmental factors, are shown in Table 4.

**Table 4.** Consumer factors and system environmental factors—path coefficient results table.

| Path | Estimate | B | S.E | C.R | ρ-Value | Sig. |
|---|---|---|---|---|---|---|
| H1: $X_{11} \to Y$ | 0.043 | 0.083 | 0.034 | 1.285 | 0.199 | No |
| H2: $X_{12} \to Y$ | 0.121 | 0.333 | 0.021 | 5.810 | *** | Yes |
| H3: $X_{13} \to Y$ | 0.202 | 0.520 | 0.027 | 7.564 | *** | Yes |
| H4: $X_{21} \to Y$ | −0.079 | 0.151 | 0.054 | 1.465 | 0.143 | No |
| H5: $X_{22} \to Y$ | 0.101 | 0.229 | 0.043 | 2.337 | * | Yes |
| H6: $X_{23} \to Y$ | 0.508 | 0.843 | 0.085 | 5.979 | *** | Yes |

Note: In the ρ-value column, *** indicates $p < 0.001$ and * indicates $p < 0.05$.

In the above table, Estimate is the path coefficient, B marks the standardized path coefficient, S.E is the standard error of the regression sampling distribution and C.R and ρ-value are used to measure significance: less than 1.96 means it failed the significance test; C.R greater than 1.96, i.e., $p < 0.05$, means that the path coefficient is significant at the level of 0.05; C.R greater than 2.58, i.e., $p < 0.01$, represents that the path coefficient is significant at the level of 0.01; and C.R greater than 3.3, i.e., $p < 0.001$, represents that the path is significant at the level of 0.001. According to the results of path analysis, it can be found that personal vehicle knowledge level, personal online shopping experience, third-party certification and industry technology level have a significant positive influence on second-hand car e-commerce platform consumer trust; personal inherent trust tendency and laws and regulations cannot positively promote second-hand car e-commerce platform consumer trust.

### 3.2.2. Linear Regression Analysis

The relationship between the remaining two groups of variables and consumer trust was investigated using regression analysis.

1. Regression analysis of website/APP factors (X3) and consumer trust (Y)

Two sub-variables under the website/APP factor (X3), ease of use of the website/APP (X31) and security of the website/APP (X32), were regressed on the dependent variable consumer trust. The regression results are presented in the Table 5.

**Table 5.** Regression results of website/APP factors.

| Variables | Non-Standardized Coefficients | Standardized Coefficients | t | Sig. | VIF |
|---|---|---|---|---|---|
| Constant | 5.272 | | 16.992 | 0.000 | |
| $X_{31}$ | 0.318 | 0.468 | 9.526 | 0.000 | 2.186 |
| $X_{32}$ | 0.231 | 0.328 | 6.676 | 0.000 | |
| | Sig.F | | | Sig. = 0.000 | |
| | $R^2$ | | | 0.552 | |
| | Adjusted $R^2$ | | | 0.549 | |
| | Durbin–Watson coefficient | | | 1.826 | |

In the regression for the website/APP factor, the adjusted $R^2$ value of 0.549 is greater than 0.4, indicating that the ease of use of the website/APP (X31) and the security of the website/APP (X32) contribute to 54.9% of the degree of change in consumer trust (Y). The significance of the F-value in the regression results is under 0.05, which shows that the model has a good goodness of fit. Looking at the Durbin–Watson coefficient and VIF values, after the regression, the Durbin–Watson coefficient of ease of use of the website/APP (X31) and security of website/APP (X32) is 1.826, which is close to 2, and the VIF value is 2.186, which is in the 1–10 acceptable range. Based on the above two points, it can be concluded that the ease of use of the website/APP (X31) and the security of the website/APP (X32) do not have first-order autocorrelation and multicollinearity, and can support linear regression analysis.

The values of the standard coefficients of ease of use of the website/APP(X31) and the security of the website/APP (X32) are 0.468 and 0.328, respectively, indicating that they both have a positive effect on consumer trust, and that ease of use of the website/APP (X31) has the largest effect on consumer trust (Y). The significance of both variables is below 0.05, which shows that ease of use of the website/APP (X31) and the security of the website/APP (X32) have a significant positive relationship with consumer trust. The conclusion of the regression analysis verifies hypotheses H7 and H8. The standardized regression coefficients were used to establish the equation:

Consumer trust = 0.468 × ease of use of the website/APP (X31) + 0.328 × security of the website/APP (X32).

2. Regression analysis of platform company factors (X4) and consumer trust (Y)

Multiple regression analysis was conducted on the two sub-variables under the platform company factors (X4), company size (X41) and company brand reputation (X42), and the dependent variable consumer trust, and the regression analysis results are tabulated as Table 6.

**Table 6.** Regression results of platform company factors.

| Variables | Non-Standardized Coefficients | Standardized Coefficients | t | Sig. | VIF |
|---|---|---|---|---|---|
| Constant | 5.312 | | 13.064 | 0.000 | |
| $X_{41}$ | 0.221 | 0.312 | 5.232 | 0.000 | 2.465 |
| $X_{42}$ | 0.293 | 0.372 | 6.247 | 0.000 | |
| | Sig.F | | | Sig. = 0.000 | |
| | $R^2$ | | | 0.414 | |
| | Adjusted $R^2$ | | | 0.411 | |
| | Durbin–Watson coefficient | | | 2.013 | |

In the regression analysis of the platform company factors, the value of adjusted $R^2$ is 0.411, which is greater than 0.4. The company size (X41) and company brand reputation (X42) lead to a 41.1% degree of change in consumer trust, and the significance of the F-value in the regression results is under 0.05, which shows that the model has a good goodness of fit. Looking at the Durbin–Watson coefficient and VIF values, the Durbin–Watson coefficient of the company size (X41) and company brand reputation (X42) is 2.013, which is close to 2, and the VIF value is 2.465, which is between 1 and 10 in the acceptable range. Based on the above two points, it can be concluded that there is no first-order autocorrelation and multicollinearity between company size (X41) and company brand reputation (X42), which can support linear regression analysis.

The values of the standard coefficients of company size (X41) and company brand reputation (X42) are 0.312 and 0.372, respectively, indicating that they both have a positive effect on consumer trust, and that company brand reputation has the largest effect on consumer trust. The significance of t-values for both variables is below 0.05, which shows that company size (X41) and company brand reputation (X42) have a significant positive relationship with consumer trust. Hypotheses H9 and H10 pass the test. The standardized regression coefficients were used to establish the regression equation.

Consumer trust = 0.312 × company size (X41) + 0.372 × company brand reputation (X42).

*3.3. Results Summary*

A total of 10 hypotheses have been proposed in this paper. Except H1 and H4, all the other hypotheses have passed the verification. The explanation is as follows:

(1) The influence coefficient of consumers' personal trust tendency on the trust standardization path of a second-hand car e-commerce platform is 0.083, and the ρ-value is 0.199, which is greater than the minimum requirement of 0.05 for the significance test and fails to pass the significance test. Hypothesis H1 is not valid. This may be because people are more cautious about goods such as cars on virtual trading

platforms. Consumers' old trust mechanisms cannot be transferred to something as new as online second-hand car trading.

(2) The influence coefficient of consumers' personal vehicle knowledge on the standardized path of trust of a second-hand car e-commerce platform is 0.333, and the $\rho$-value is less than 0.001, passing the significance test, and the influence degree is 0.333, assuming H2 is valid. Its internal mechanism is as follows: people who know more about vehicle parameters will be more capable and confident to judge the quality of second-hand cars, so they are more likely to buy on second-hand car e-commerce platforms.

(3) The influence coefficient of consumers' personal online shopping experience on the standardized path of trust of a second-hand car e-commerce platform is 0.520, and the $\rho$-value is less than 0.001, passing the significance test, and the influence degree is 0.333. Hypothesis H3 is valid. A second-hand car e-commerce platform is essentially similar to online trading platforms for other commodities. In addition, a second-hand car e-commerce platform will design the transaction process and services in line with the habits of online shoppers. Therefore, consumers with more online shopping experience will have lower psychological barriers to a second-hand car e-commerce platform and will be more likely to have purchasing behaviors.

(4) The influence coefficient of laws and regulations on the standardized path of trust of a second-hand car e-commerce platform is −0.079, and the $\rho$-value is 0.143, which is greater than the minimum requirement of 0.05, and fails to pass the significance test. Hypothesis H4 is not valid. The reason may be that Chinese e-commerce law is not perfect, and it takes some time and process to popularize the law.

(5) The influence coefficient of the third-party certification on the standardized path of trust of a second-hand car e-commerce platform is 0.229, and the $\rho$-value is less than 0.01. It passes the significance test, and the influence degree is 0.229, assuming that H5 is valid. Third-party certification agencies can improve the trust in a second-hand car e-commerce platform in the eyes of consumers. The more authoritative the certification, the stronger the sense of trust.

(6) The influence coefficient of industrial technology on the trust standardization path of a second-hand car e-commerce platform is 0.508, and the $\rho$-value is less than 0.01. It passes the significance test, and the influence degree is 0.508. Hypothesis H6 is valid. The more technologically sophisticated the industry, the more trustworthy the commodity itself and the transaction itself. On the contrary, new energy vehicles are rarely traded on second-hand car trading platforms because their technology is not mature.

(7) The trust standard coefficient of perceived ease of use of the website/APP of a second-hand car e-commerce platform is 0.468, the $\rho$-value is less than 0.001, passing the significance test, and the influence degree is 0.508, assuming that H7 is valid. If the interaction design of a platform's website/APP allows consumers to learn quickly and find the information they want easily, it will be easier to build trust.

(8) The standard coefficient of trust of the perceived safety of the website/APP of a second-hand car e-commerce platform is 0.328, the $\rho$-value is less than 0.001, passing the significance test, and the influence degree is 0.328, assuming that H8 is valid. Consumers receive security information such as website/APP security agreement terms and privacy protection measures. The easier this information is perceived by consumers, the stronger the sense of trust will be.

(9) The standard coefficient of trust of the scale and strength of a company's platform of a second-hand car e-commerce platform is 0.312, and the $\rho$-value is less than 0.001, passing the significance test, and the influence degree is 0.312, assuming that H9 is valid. This is consistent with the general law of business, which is that the larger the company, the more people recognize its strength. Consumers are more likely to trade with large firms than small ones.

(10) The standard coefficient of trust of a company's brand reputation of a second-hand car e-commerce platform is 0.372, and the $\rho$-value is less than 0.001, which passes

the significance test, and the influence degree is 0.372. Hypothesis H10 is valid. The better a platform's brand reputation, the easier it is to create trust. Consumers tend to pay less attention to product performance and price due to brand trust.

## 4. Conclusions and Discussions

### 4.1. Theoretical Implications

Results of this study indicate that consumer trust in used-car e-commerce comes from many factors. In terms of personal factors, consumers' mastery of commodities and familiarity with online shopping are the internal factors that determine whether consumers choose to purchase second-hand cars on a second-hand car e-commerce platform. Therefore, businesses selling vehicles on a platform should be highly professional. In terms of system factors, technical level and authoritative certification are the external factors that determine whether consumers trust used-car e-commerce. It can be considered that the more mature the automobile industry is, the more likely it is to trade second-hand cars through the Internet, and the more likely it is to succeed in the e-commerce model of second-hand cars. On the contrary, when the industry itself is not mature, it may be impossible to conduct transactions through the network. Business factors such as enterprise scale, brand influence and user interface experience can affect the formation of consumer trust. Industry facts also show that it is difficult for used-car e-commerce enterprises to fully rely on their own resources and capabilities in the early stage and gradually accumulate customers in order to achieve endogenous development. They often introduce capital at an early stage, combine industry forces to rapidly increase the flow, and then iteratively optimize their products and services to achieve endogenous development.

### 4.2. Practical Implications

A second-hand car e-commerce platform has the characteristics of high unit price and low frequency. Therefore, establishing the initial trust of consumers and improving the degree of trust of consumers is the top priority to trigger the purchase behavior [40].

Broadly speaking, in order to strengthen the environment of trust in the second-hand car trade, the following should be considered:

- Improve the Internet legal system and standardize the trading order of the second-hand car e-commerce platform.
- Increase publicity of legal knowledge to improve consumers' cognition of second-hand car e-commerce trading rules.

In a narrow sense, a second-hand car e-commerce platform should improve its service management level, improve consumers' sense of experience on the service contact surface and thus increase the sense of trust. Specific measures are as follows:

- A platform should take consumer groups who prefer online shopping as the preferred target market.
- The selection and evaluation mechanism of service providers should be designed for the second-hand car e-commerce platform to ensure the service level of service providers.
- Introduce local third-party authoritative certifications, including second-hand car quality certification, e-commerce platform qualification certification, etc.
- The interface design of the platform should use colors and layouts that help consumers improve their trust and highlight information such as quality certification and transaction evaluation.
- Attach importance to brand image construction and maintenance, and constantly improve brand awareness and reputation.
- A second-hand car e-commerce platform should train the service team and improve the professional quality of the team.

*4.3. Limitations and Future Research*

This study is not free of limitations. First of all, some of the respondents used are mainly from the researcher's circle of friends, which may have some limitations. In the future, user feedback data can be obtained through more channels to improve sample quality. Second, this study only uses the questionnaire method, not the qualitative method, so the information obtained is still limited. Some representative used-car e-commerce enterprises can be selected to carry out in-depth case studies.

**Author Contributions:** Conceptualization, X.L. and J.M.; Methodology, X.L. and J.M.; Validation, J.M.; Investigation, X.L., J.M., X.Z. and R.Y.; Writing—original draft, X.L., J.M. and X.Z.; Writing—review & editing, J.M., X.Z. and R.Y.; Project administration, X.L. All authors have read and agreed to the published version of the manuscript.

**Funding:** This research received no external funding.

**Institutional Review Board Statement:** Not applicable.

**Informed Consent Statement:** Not applicable.

**Data Availability Statement:** Not applicable.

**Conflicts of Interest:** The authors declare no conflict of interest.

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
