# Peer review of "Research on Consumer Trust Mechanism in China’s B2C E-Commerce Platform for Second-Hand Cars"

_sustainability, doi:10.3390/su15054244_

Round 1

Reviewer 1 Report

Thanks for submitting the article. It looks to be an interesting topic. Nevertheless, it lacks the justification for researching this topic. I struggled to understand what the gap is in the literature. The gap should be based on previous papers in the field. It looks like a good case study but needs to be dressed in academic literature. It is a very contextual study rather than an academic one.

Do you have ethical approval for collecting the data? Please state it.

Also, I was wondering what the exact theoretical and practical implications of this research are, the limitations, and how future research should be done in this field. 

Author Response

Thank you for your comments. Your comments remind us that we need to focus more on academic writing. Therefore, in the first part, we make a supplementary analysis of the literature; In the last part, from the perspective of theory and practice, especially the management enlightenment of enterprises, the author summarizes and explains the limitations of the study. In the uploaded revision file, we made detailed modifications. Please check.

Thank you very much!

In addition, when we distributed the questionnaire, we made a detailed explanation to the respondents and obtained their consent. This study is only about the subjective cognition of the interviewees, and does not involve privacy issues such as their personal information. Therefore, the data we collect is legal.

Reviewer 2 Report

This is a relevant topic for both academia and management. The work was well developed and there are only a few aspects that can be improved before a possible publication.

It is necessary for Figure 2 to include each number of hypotheses associated with each variable, since there are many hypotheses to test and the inclusion will make it easier for readers to visually associate them. 

Include the scales used and/or an explanation in some detail of how the constructs and/or variables were chosen and measured. This will make it easier for other researchers, if they wish, to replicate the research or advance the line of knowledge.

Review that common method bias is not present.

It is a good idea to have presented a summary of the results; it helps to clearly situate the research findings.

It is necessary to consider a discussion section. It not only links the results to the theory, but takes the authors and readers to a deeper level of understanding of the findings and implications.

Review to include key limitations and suggestions for future research.

Author Response

First of all, thank you very much for your comments and suggestions. We have made many modifications to the article, and the brief introduction is as follows:

1. According to your suggestion, we have added hypothetical tags in Figure 2.

2. In section 2.2, we added a table to show the measurement method for each variable.

3. In the last part, it summarizes the theoretical research conclusions of this paper. For the laws found in the study, the paper puts forward strategic suggestions from the perspective of enterprise management. Finally, the limitations of the study are stated.

4. In addition, many other places have been revised, such as the first part.

For details, please check the uploaded revision file.

Thank you very much!

Reviewer 3 Report

The research presented is interesting and highly relevant. The paper is well written and the ideas are clearly exposed, as well as the methodologies adopted. Although the survey is not representative of the Chinese population, the sample obtained for relevance has the necessary credibility. All information was presented with the necessary transparency.

There are only 3 details that should be reviewed and improved:

- Neither in the title nor in the Abstract is there any reference that the study is located in China. This reference has to be clear at least in the Abstract.

- The literature review is good, but it should be complemented with some more recent scientific references in order to demonstrate the timeliness of the study.

- In the Conclusions chapter, the authors should present recommendations for future study.

Author Response

First of all, thank you very much for your comments. According to your comments, we have made a comprehensive and detailed revision to the full text. Please refer to the revised document in the attachment for details.

1. In the title, abstract and previous paragraphs of the paper, some words are added, indicating that this study is aimed at the Chinese market.

2. The first part adds some scientific literature, especially on consumer trust.

3. In the last part, it summarizes the conclusions of the full text, and explains the limitations of the article and the corresponding improvement direction.

Reviewer 4 Report

The authors  explored and analyzed the influencing factors of consumer trust.

The importance and soundness of the proposed hypotheses- the established hypotheses were correctly documented, established and tested.

Are there sufficient details given to replicate the proposed experimental procedures and analysis? I suggest the authors to add the content of the questionnaire in the Appendix section.

Are there sufficient outcome-neutral tests of the hypotheses, including positive controls and quality checks? Yes, the authors considered sufficient quality checks.

I would suggest the authors to compare the results of their study with other results in the field and to highlight specific contributions of their research.

Author Response

First of all, thank you very much for your comments and suggestions. We have made necessary and detailed revisions to the full text. Please refer to the revised manuscript in the attachment.

For example:

1. In the first part, we added some scientific literature analysis to illustrate the theoretical contribution of our research and further let readers understand the problems we want to study.

2. In the second part, we will add more detailed descriptions on the distribution and collection of questionnaires and the measurement of each variable.

3. In addition, we believe that the empirical process and results meet the requirements of methodology.

Round 2

Reviewer 1 Report

Thanks for submitting the updated version of this paper; I can see improvements.

You have not addressed my comment about ethical approval!!

 But the Theoretical implication and contribution need more clarity: I think you should be precise on which stream of studies you are contributing to. You have already mentioned the gap in the studies in your literature review. It would be beneficial if you could restate which papers you contribute to in your theoretical implication section. 

Author Response

Thank you for your review!

According to your comments, we have revised it. For details, please refer to the attachment - modified document. Mainly, we further improved the theoretical analysis and summary.

About the ethical approval you mentioned:

We believe that this article is completely in line with academic ethics. First of all, the idea of this article comes from our own research and practice. Secondly, the research process and results are all done by us. Finally, we wrote the article.

Thank you again!

Reviewer 4 Report

The authors responded to my suggestions.

Author Response

Thank you for your review.

We have made some changes to the article again.

For details, please refer to the article in the attachment.

Round 3

Reviewer 1 Report

I am not negotiating if the article is written by you or not !! You mentioned that this was part of your research. So my question did you have ethical approval before gathering the data? State how the data is kept and saved. And how have you dealt with respondent identity? You need to shed light on these aspects within the methodology section. Clearly state that this was part of your research and state the ethical approval number if you have any. 

Author Response

Thank you for your review. 

Sorry, we misunderstood your meaning. For the questionnaire, we have already got the informed consent.

At the same time, before we asked the respondents to fill in the questionnaire, we solicited the opinions of the respondents and explained the purpose of the questionnaire in detail. The questionnaire does not involve personal risks. All the friends who responded to the valid questionnaire agreed to help us fill in the questionnaire and explore theoretical research.

The questionnaire includes introduction and body. In the introduction, we explained the purpose and significance of this study in detail, and emphasized that it is only for theoretical research, does not involve personal privacy, and does not pose any risk to individuals and the public. The respondents can refuse to fill in. In the body, they do not need to fill in the identity information.

Thank you again!